# On the Spectral Differences Between NTK and CNTK and Their Implications for Point Cloud Recognition

**Yuanqu Mou** [†]
Department of Computer Science
Nanjing University
mouyq@smail.nju.edu.cn

**Chang Gou** [†]
 Nanjing University
cgou@smail.nju.edu.cn

**Haiyang Bai**
Nanjing University
baihy@smail.nju.edu.cn

**Jia Liu** [*]
State Key Laboratory for Novel Software Technology
Nanjing University
jialiu@nju.edu.cn

## Abstract

The Convolutional Neural Tangent Kernel (CNTK) offers a principled framework for understanding convolutional architectures in the infinite-width regime. However, a comprehensive spectral comparison between CNTK and the classical Neural Tangent Kernel (NTK) remains underexplored. In this work, we present a detailed analysis of the spectral properties of CNTK and NTK, revealing that point cloud data exhibits a stronger alignment with the spectral bias of CNTK than images. This finding suggests that convolutional structures are inherently more suited to such geometric and irregular data formats. Based on this insight, we implement CNTK-based kernel regression for point cloud recognition tasks and demonstrate that it significantly outperforms NTK and other kernel baselines, especially in low-data settings. Furthermore, we derive a closed-form expression that connects CNTK with NTK in hybrid architectures. In addition, we introduce a closed-form of CNTK followed by NTK, while not the main focus, achieves strong empirical performance when applied to point-cloud tasks. Our study not only provides new theoretical understanding of spectral behaviors in neural tangent kernels but also shows that these insights can guide the practical design of CNTK-based regression for structured data such as point clouds.

## 1 Introduction

Neural Tangent Kernel (NTK) Jacot et al. (2018) emerged as a critical theoretical framework for understanding the training dynamics and generalization behaviors of deep neural networks Golikov et al. (2022). Among its variants, the Convolutional Neural Tangent Kernel (CNTK) Arora et al. (2019a) provides a kernel-based perspective on convolutional neural networks (CNNs), enabling precise analysis of their inductive biases and convergence properties. Building on this foundation, existing studies Geifman et al. (2022)Cagnetta et al. (2023)Barzilai et al. (2022)Bietti & Mairal (2019) have explored the properties of CNTK as a kernel and extended CNTK in various directions Li et al. (2019). However, there has been a lack of discussion on the differences between CNTK and NTK when applied to the same dataset. This paper aims to address this theoretical issue by investigating the following question:

---

[*]Corresponding author. [†] The two authors contribute equally to this work.

*Under a given data assumption, what are the spectral differences between NTK and CNTK?*

To address this issue, it is first necessary to consider an appropriate data distribution. Existing studies often assume that data is uniformly distributed on a high-dimensional sphere, without taking into account the inherent tensor structure of the data. In this paper, we assume that the $d+1$-dimensional data is distributed with any distribution over the $d$-dimensional tensor space (with a generally non-diagonal covariance matrix), and is insignificant across the channel dimension (though not necessarily independent). Under this assumption, we identify two major differences between NTK and CNTK when evaluated on datasets drawn from the same distribution:

(I) For any $L > 0$, the mean of the eigenvalues $m_K$ under $L$-layer NTK is greater than that under $L$-layer CNTK, indicating that NTK tends to have larger eigenvalues on average.

(II) For any $L, L' > 0$, the spread $\beta_K$ of the eigenvalue distribution under any $L$-layer NTK is smaller than that under any $L'$-layer CNTK, suggesting that NTK exhibits a more concentrated spectrum compared to CNTK.

Observation (II) explains why convolutional networks generally achieve better generalization than fully connected networks since the generalization risk can be bounded by the varianceBartlett et al. (2020), while (I) provides a theoretical justification for the empirical design choice of adding a few MLP layers after pooling in deep convolutional networks.

Based on this finding, this paper addresses a practical application problem:

*How do MLPs (NTK) and convolutional networks (CNTK) differ in their influence on data with different levels of relevance?*

To address this issue, we marked the metric $\beta_K$ aligned with the suitability of the convolutional structure for data analysis. By fitting a dataset to our assumed distribution, $\beta_K$ quantifies the degree to which NTK exhibits a more concentrated spectrum compared to CNTK, as described in observation (II). This value serves as an indicator of how differently MLPs and convolutional networks are expected to perform on the given data.

In addition, most existing studies on CNTK are limited to the commonly used two-dimensional case, with little discussion on more general dimensional settings. Therefore, this paper extends the analysis to convolutional kernels of arbitrary dimension. Furthermore, we analyzed the propagation of CNTK-NTK kernel computations in convolutional networks followed by nonlinear fully connected layers, which is an empirical setup. In this work, such a setup can be interpreted as leveraging the broad spectrum of CNTK eigenvalues (II) and the bias of NTK towards larger shifts (I).

Specifically, as a case study, we find that point cloud tasks rely more heavily on convolutional structures compared to image tasks. Motivated by this, we further investigate kernel regression based on one-dimensional convolutions in the context of point cloud tasks.

## 1.1 CONTRIBUTION

We considered the notation and framework of CNTK under general dimensions that can be connected to NTK. Within this framework, we revealed the spectral differences between NTK and CNTK that are distribution-free. Furthermore, based on the theoretical findings, we proposed the concept of 'Convolutional Suitability' for data analysis, which measures how suitable a given dataset is for convolutional networks. Finally, we applied CNTK to point cloud tasks for the first time, including kernel regression experiments based on CNTK and the application of our theoretical conclusions to point cloud datasets.

## 1.2 NOTATION

We assume that each point $x_i$ comes from a $d$-order tensor space $\mathbb{R}^{\Pi_{i=1}^d h_i}$, where $h_i$ denotes the length along the $i$-th dimension. Each sample has $n_0$ channels of such point, forming a data point $x = \{x_1, x_2, \ldots, x_{n_0}\}$. For such tensor data, we do not use boldface notation. In this paper, only data constructed from $\mathbb{R}^{\Pi_{i=1}^d h_i} \times \mathbb{R}^{\Pi_{i=1}^d h_i}$ are written in bold to distinguish them. Define the index set $p, q \in [[h_i]_d]$, where $[h_i] = \{1, 2, \ldots, h_i\}$, and $[[h_i]_d] = [h_1] \times [h_2] \times \cdots \times [h_d]$, with $\times$ denoting the Cartesian product. The length of the convolutional kernel is defined as $s$ (For simplicity, we assume

that the convolutional kernel is square.) $N(p)$ is denoted as convolution kernel indices centered at $p$, that is, $N(p) = \{[p_1 - \lfloor \frac{s}{2} \rfloor, p_1 + \lfloor \frac{s}{2} \rfloor], ..., [p_d - \lfloor \frac{s}{2} \rfloor, p_d + \lfloor \frac{s}{2} \rfloor]\}$.

For an activation function $\phi(x)$, we denote its derivative as $\dot{\phi}(x)$, and its dual function with respect to a distribution $D$ as $\tilde{\phi}(D)_{x,x' \sim D} = \mathbb{E}[\phi(x)\phi(x')]$. We define $\odot$ as the Schur (element-wise) product, $\otimes$ as the Kronecker (tensor) product, and $\cdot$ as scalar multiplication or standard matrix–matrix multiplication.

## 2 CONVOLUTIONAL NEURAL TANGENT KERNEL

In this section, we provide a brief introduction to the Convolutional Neural Tangent Kernel (CNTK). For the purpose of subsequent generalization, the definition is given in the context of inputs with arbitrary dimensions , not restricted to the two-dimensional case. Following the recursion in Jacot et al. (2018) Arora et al. (2019a), CNTK for unified CNNs is defined as follows.

**Definition 1.** *Given two samples $x, x' \in \mathbb{R}^{\Pi_{i=1}^d h_i \times n_0}$, let $p, q$ be a position in $[[h_i]_d]$ with neighborhood $N(p)$. Define the **covariance matrix**, **0-order expectation** and **1-order expectation** of $f_p^{(l)}(x)$ and $f_q^{(l)}(x')$ as follows:*

$$\mathbf{\Lambda}_{p,q}^{(l)}(x,x') = \begin{pmatrix} \mathcal{E}_{p,q}^{(l-1)}(x,x) & \mathcal{E}_{p,q}^{(l-1)}(x,x') \\ \mathcal{E}_{p,q}^{(l-1)}(x',x) & \mathcal{E}_{p,q}^{(l-1)}(x',x') \end{pmatrix},$$

$$\mathbf{\Sigma}_{p,q}^{(l)}(x,x') = \mathbb{E}_{(u,v) \sim \mathcal{N}\left(\mathbf{0},\mathbf{\Lambda}_{p,q}^{(l)}(x,x')\right)} \left[ \phi(u)\phi(v) \right], \tag{1}$$

$$\dot{\mathbf{\Sigma}}_{p,q}^{(l)}(x,x') = \mathbb{E}_{(u,v) \sim \mathcal{N}\left(\mathbf{0},\mathbf{\Lambda}_{p,q}^{(l)}(x,x')\right)} \left[ \dot{\phi}(u)\dot{\phi}(v) \right],$$

*with recursion for $\mathcal{E}_{p,q}^{(l)}(x,x')$ is:*

$$\mathcal{E}_{p,q}^{(l)}(x,x') = tr\left( \mathbf{\Sigma}_{N(p),N(q)}^{(l)}(x,x') \right). \tag{2}$$

*Define the trace of a tensor $\mathbf{T} \in \mathbb{R}^{\Pi_{i=1}^d h_i \times \Pi_{i=1}^d h_i}$ as $tr(\mathbf{T}) = \sum_{p \in [[h_i]_d]} \mathbf{T}_{p,p}$. Recursive foundation is:*

$$\mathbf{\Sigma}_{p,q}^{*(0)}(x,x') = x \otimes x'. \tag{3}$$

When the convolution dimension is $d$, there will also be interactions between the elements within the tensors at $l$-layer's neurons. For any two samples $x, x'$, this relationship is described by the three tensors $\mathbf{\Lambda}^{(l)}(x,x') \in \mathbb{R}^{\Pi_{i=1}^d h_i \times \Pi_{i=1}^d h_i \times 2 \times 2}$, $\mathbf{\Sigma}^{(l)}(x,x'), \dot{\mathbf{\Sigma}}^{(l)}(x,x') \in \mathbb{R}^{\Pi_{i=1}^d h_i \times \Pi_{i=1}^d h_i}$ in the NTK field, as mentioned above. Consequently, we can derive the specific recursive formula for CNTK.

**Lemma 1.** *Given two samples $x, x' \in \mathbb{R}^{\Pi_{i=1}^d h_i \times n_0}$, and position $p$ with its neighbourhood, $N(p)$. The CNTK without pooling is computed as:*

$$K_{wop}(x,x') = tr\left( \mathbf{K}^{(L)}(x,x'), \right) \tag{4}$$

*the CNTK with the global average pooling $\mathcal{P}_{av}$ is computed as:*

$$K_{wp}(x,x') = \mathcal{P}_{av}\left( \mathbf{K}^{(L)}(x,x') \right), \tag{5}$$

*in which $\mathbf{K}^{(l)}(x,x')$ is $l$-**layer tensor kernel** with recursion for $1 \le l \le L-1$ and $p$-entry as:*

$$K_{p,q}^{(l)}(x,x') = tr\left( K_{N(p),N(q)}^{(l-1)} \odot \dot{\mathbf{\Sigma}}_{N(p),N(q)}^{(l-1)}(x,x') \right) + \mathcal{E}_{p,q}^{(l-1)}(x,x'). \tag{6}$$

The derivation details are provided in the original researchArora et al. (2019a). Nevertheless, we also provide a derivation in the general-dimensional case in Appendix B. Note that when we consider the case $d = 0$, corresponding to the structure of MLPs, we have $p = q$ fixed and $N(p) = p$, which recovers the standard NTK formulation. By default, we consider the CNTK with pooling layers included in the following.

## 3 CNTK FOLLOWED BY NTK

While the Convolutional Neural Tangent Kernel (CNTK) captures the infinite-width behavior of vanilla convolutional networks, it does not account for architectures commonly used in practice, where a shallow MLP is typically appended after the convolutional backbone. In this section, we derive a closed-form expression of the CNTK for a more general architecture: a $d_1$-dimensional convolutional network followed by a $d_2$-dimensional convolutional (or fully connected when $d_2 = 0$) network.

Generally, let the network topology be determined by two sets: $\{L_1, L_2\}$ and $\{d_1, d_2\}$, where $d_1 > d_2$. This represents a network consisting of $L_1$ layers of $d_1$-dimensional convolutional layers and $L_2$ layers of $d_2$-dimensional convolutional layers, connected by an average pooling layer. We can then propose the combined dimension CNTK formally as follows:

**Proposition 1.** *Suppose $d_1$-dimension sub convolutional neural network $\boldsymbol{f}_1 : \mathbb{R}^{\Pi_{i=1}^{d_1} h_i^1 \times n_0^1} \to \mathbb{R}^{\Pi_{i=1}^{d} h_i^1 \times n_{L_1}^1}$ and $d_2$-dimension sub convolutional neural network $\boldsymbol{f}_2 : \mathbb{R}^{\Pi_{i=1}^{d_2} h_i^2 \times n_0^2} \to \mathbb{R}^{\Pi_{i=1}^{d} h_i^2 \times n_{L_2}^2}$ with $d_1 > d_2$, $n_{L_1}^1 = n_0^2$ and $h_i^1 = h_i^2$ for $i \in [d_2]$, generalizing convolutional neural network $\boldsymbol{f}_G : \mathbb{R}^{\Pi_{i=1}^{d_1} h_i^1 \times n_0^1} \to \mathbb{R}^{\Pi_{i=1}^{d} h_i^1 \times n_{L_2}^2}$ is defined as:*

$$\boldsymbol{f}_G(x) = \boldsymbol{f}_2 \left( \mathcal{P}_{av}^{d_1 \to d_2} \left( \boldsymbol{f}_1(x) \right) \right), \tag{7}$$

*in which $\mathcal{P}_{av}^{d_1 \to d_2} : \mathbb{R}^{\Pi_{i=1}^{d} h_i^1 \times n_{L_1}^1} \to \mathbb{R}^{\Pi_{i=1}^{d} h_i^2 \times n_{L_2}^2}$ is local average pooling ,when $\boldsymbol{f}_c$ is standard initialization and in which, $n_1^1, n_2^1, ..., n_{L_1-1}^1, n_1^2, ..., n_{L_2-1}^2 \to \infty$, its corresponding kernel is:*

$$K_c(x, x') = \mathcal{P}_{av} \left( K^{(L_1+L_2)}(x, x') \right), \tag{8}$$

*at the point of connection, for any samples $x, x'$:*

$$\Sigma^{(L_1+1)}(x, x') = \mathcal{P}_{av}^{d_1 \to d_2} \left( \Sigma^{(L_1)}(x, x') \right). \tag{9}$$

In fact, such a structure can be applied more than once. In practical scenarios, convolutional networks are typically connected to only a few MLP layers, and therefore we usually have $d_2 = 0$, which is named CNTK-NTK in the following.

## 4 SPECTRAL COMPARISON

In this section, we formally present our results about the spectral properties of CNTK and NTK. For a $d$-dimensional tensor space, we consider data sampled from $\mathbb{R}^{\Pi_{i=1}^{d} h_i \times n_0}$ and following a $H$-dimensional distribution (can be any distribution) with covariance matrix $\boldsymbol{\sigma}_H$ in which $H = \Pi_{i=1}^{d} h_i$. We consider convolutional networks with kernel size 1 and average pooling layers.

### 4.1 THEORETICAL RESULTS

Let $K_{CNTK}^{(L)}$ denote the CNTK of an $L$-layer convolutional neural network, and $K_{NTK}^{(L)}$ denote the NTK of an $L$-layer MLP. We define the statistics about eigenvalues of kernel matrix $K$ as: $m_K^{(L)} = \frac{1}{N} tr(K^{(L)})$, $s_K = \frac{1}{N^2} tr(K^{(L)} K^{(L)\top})$ and $\beta_K^{(L)} = \frac{s_K^{(L)}}{(m_K^{(L)})^2}$. These three quantities reflect certain statistical characteristics of the eigenvalues of the kernel matrix: $m_K$ denotes the mean of the eigenvalues, $s_K$ represents the mean of the squared eigenvalues, and $\beta_K$ indicates the dispersion of the eigenvalues. Formally, based on results from matrix analysis literature Cai et al. (2024)Horn & Johnson (2012), we have:

**Lemma 2.** *Given a $N \times N$ positive definite matrix $K$, let its set of eigenvalues be denoted as $\lambda_1 \leq \lambda_2 \leq ... \leq \lambda_N$, then:*

$$m_K = \frac{1}{N} \sum_{i=1}^{N} \lambda_i, \quad s_K = \frac{1}{N^2} \sum_{i=1}^{N} \lambda_i^2, \quad \frac{\lambda_{max}}{\lambda_{min}} \geq N \beta_K. \tag{10}$$

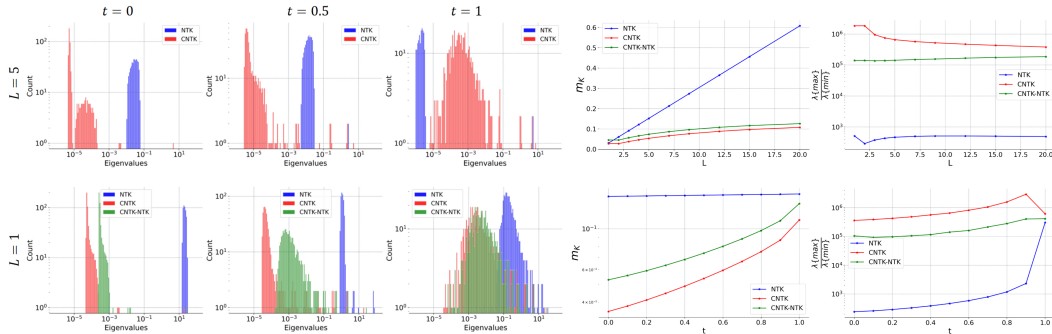

Figure 1: Experiments on synthetic data, $t$ denotes the correlation of the multidimensional Gaussian distribution $g$, and $N$ refers to the Gaussian distribution $\mathcal{N}$. $L$ represents the number of network layers, where $L = 1$ corresponds to the neural tangent kernel $K^{(0)}$ without passing through any network. In the right figure: in the upper part, $t$ is always set to 0.5; in the lower part, $L$ is fixed to 5. 'CNTK-NTK' refers to the CNTK of depth $L$ followed by an additional two-layer MLP.

Furthermore, $1/\beta_K$ is essentially the same as the quantity *effective rank* defined in Bartlett et al. (2020), which was shown to upper-bound the generalization error of kernel regression. In our analysis, we use $\beta_K$ as a comparative metric to assess how NTK and CNTK differ in their generalization ability on data drawn from arbitrary distributions, thereby shedding light on what kinds of data benefit more from convolutional structures. In addition, $m_K$, as an offset term of the kernel matrix, reflects the convergence capacity of the network. Therefore, we also include a discussion on this aspect.

**Theorem 2.** *Suppose that the activation function is Relu. When $N \to \infty$, the following holds:*

$$m_{NTK}^{(L)} \geq m_{CNTK}^{(L)}. \tag{11}$$

**Remark 1.** *For this conclusion, we do not require the data to come from any specific distribution, nor do we assume that $N$ tends to infinity. In addition, the activation function doesn't need to be ReLU; it is sufficient that the dual of the activation function is non-convex with respect to the diagonal entries. Therefore, the conclusion is general. In fact, a relatively strong statement is used in the proof process—namely, for any $x \in \mathbb{R}^{\Pi_{i=1}^{d} h_i \times n_0}$, we have $K_{NTK}^{(L)}(x,x) \geq K_{CNTK}^{(L)}(x,x)$.*

**Theorem 3.** *For any $L$-layer CNTK and $L'$-layer CNTK, under any distribution of $x$ and any activation function. When $N \to \infty$, the following holds:*

$$\beta_{NTK}^{(L)} \leq \beta_{CNTK}^{(L')} = 1. \tag{12}$$

**Remark 2.** *In fact, we can prove that for CNTK, $\beta_{CNTK}^{(L')} = 1$ holds for any layer $L'$, while for NTK, $\beta_{NTK}^{(L')} < 1$ for every layer. This follows purely from the positive semi-definiteness of the covariance matrix under random initialization, and thus holds universally without any assumption on the data distribution.*

In the analysis of NTK, the magnitude of eigenvalues affects the convergence speed of neural networks Du et al. (2018). The value of $m_K$ can reflect the convergence speed of the network. Meanwhile, $\beta_K$ indicates the spread of the eigenvalues of the kernel function; a small eigenvalue range suggests that the matrix is overly diagonalized, and Bartlett et al. (2020) has shown that overly diagonalized kernel matrices often indicate poor generalization ability. The above theorem explains why convolutional networks tend to generalize better than MLPs, and why appending fully connected layers after deep convolutional networks can accelerate training while maintaining generalization performance.

**Proposition 4.** *Suppose that covariance matrix of $x$ is $\boldsymbol{\sigma}_H$, when $N \to \infty$:*

$$m_{NTK}^{(0)} = mean(diag(\boldsymbol{\sigma}_H)), \quad m_{CNTK}^{(0)} = mean(\boldsymbol{\sigma}_H) \tag{13}$$

$$s_{NTK}^{(0)} = mean(\boldsymbol{\sigma}_H \odot \boldsymbol{\sigma}_H), \quad s_{CNTK}^{(0)} = mean(\boldsymbol{\sigma}_H \otimes \boldsymbol{\sigma}_H) \tag{14}$$

$$\beta_{NTK}^{(0)} = \frac{mean(\boldsymbol{\sigma}_H \odot \boldsymbol{\sigma}_H)}{mean(diag(\boldsymbol{\sigma}_H) \otimes diag(\boldsymbol{\sigma}_H))}, \quad \beta_{CNTK}^{(0)} = 1. \tag{15}$$

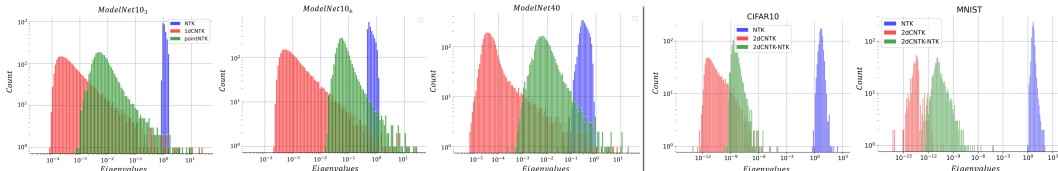

Figure 2: Histograms of Eigenvalues on:(a). Point cloud datasets ModelNet10$_3$, ModelNet10$_6$ and ModelNet40; (b) Image datasets CIFAR10 and MNIST.

Table 1: Kernel regression experiments conducted on both the point cloud and image datasets, 1dCNTK is applied to the point cloud dataset, while 2dCNTK is utilized on the image dataset. For all experiments, we set the number of layers to $L = 7$.

| Datasets | ModelNet10$_3$ | ModelNet10$_6$ | ModelNet40 | MNIST | CIFAR10 |
|---|---|---|---|---|---|
| NTK | 17.58 | 16.19 | 11.10 | 94.00 | 59.19 |
| 1dCNTK(2dCNTK) | 76.65 | 91.96 | 60.62 | 96.00 | 76.79 |
| Improvement | 59.07 | 75.77 | 49.52 | 2.00 | 17.60 |
| $\beta_K$ | 1.25e-3 | 1.12e-3 | 1.53e-3 | 3.38e-2 | 1.37e-1 |

*It is obvious that $m_{NTK}^{(0)} \geq m_{CNTK}^{(0)}$, $s_{NTK}^{(0)} \geq s_{CNTK}^{(0)}$ and $\beta_{NTK}^{(0)} \leq \beta_{CNTK}^{(0)}$. The equality of all the formulas holds if and only if $\boldsymbol{\sigma}_H$ is a constant matrix.*

This proposition suggests that the shift and distribution of CNTK and NTK eigenvalues can be described by the covariance matrix defined over the tensor input. Although the proposition formally addresses only the initial layer, our simulation experiments indicate that for CNTK and NTK with the same depth (see Figure 1), their spectral characteristics can still be effectively captured by the covariance structure of the data.

Based on this proposition, for data sampled from the same distribution, the spectral distribution of the NTK eigenvalues is associated with its covariance matrix. We define $\beta_{NTK}^{(0)}$ as a measure of the MLP fitness of the data, quantifying how well the data aligns with an MLP architecture. In the context of the tensor data considered in this paper, $\beta_{CNTK}^{(L)} - \beta_{NTK}^{(0)} = 1 - \beta_{NTK}^{(0)}$ can also be interpreted as an indicator of how suitable the data is for convolutional networks (i.e. Convolutional Suitability).

In our simulation, we use data sampled from a Gaussian distribution to analyze the spectral properties of the NTK and CNTK matrices, illustrated in Figure 1. The parameter $t$ controls the global correlation of variables, with the covariance matrix given by $\boldsymbol{\sigma} = ts\boldsymbol{C} + (1-t)s\boldsymbol{I}$ with a unit diagonal matrix $\boldsymbol{I}$ and unit constant matrix $\boldsymbol{C}$ of scale $s = 0.1$. The observed results align precisely with our theoretical predictions.

## 4.2 IN REAL SCENARIOS

As an application of the invariance of $\beta_K$ under CNTK and its relevance to NTK, we use $\beta_K^{(0)}$ as an indicator to analyze a given dataset. Assuming the data is drawn from a Gaussian distribution, we conduct kernel regression fitting for both point cloud recognition (with 1D convolutions) and image classification (with 2D convolutions). As shown in Figure 2 and Table 1, the spectral gap between CNTK and NTK is more pronounced in point cloud tasks than in image datasets. The table further indicates that point cloud tasks intrinsically benefit more from convolutional models—both theoretically, as suggested by smaller values of $\beta_K$, and empirically, as confirmed by kernel regression experiments. The inherent unorderedness of point clouds results in weak inter-sample correlations within the tensor space, which in turn yields a near-diagonal covariance structure and thus a lower $\beta_K$. Conversely, image data typically exhibits stronger structural correlations, leading to higher values of $\beta_K$. This observation implies that convolutional architectures are particularly crucial for effectively modeling point cloud data.

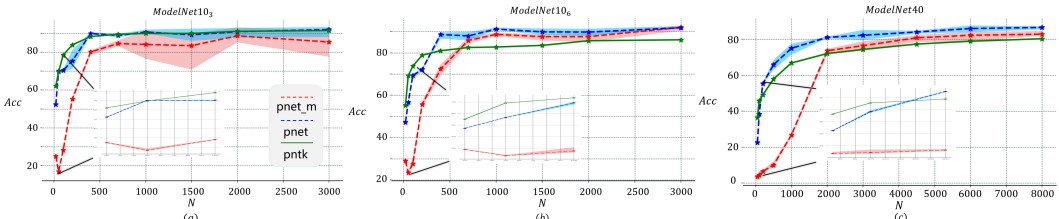

Figure 3: In the visualizations, 'pnet' denotes PointNet, 'pnet$_m$' represents PointNet$_m$, and 'pntk' corresponds to PointNTK. Supervised training results are represented with dashed lines. For each dataset, five independent experiments were conducted, and the mean results are reported. The shaded regions indicate the range of outcomes from these five experiments. Each dataset's plot features a subfigure in the lower-left corner, showing results for training with a small sample size, and a subfigure in the lower-right corner, illustrating results for training with a large sample size.

## 5 KERNEL REGRESSION FOR POINT CLOUD RECOGNITION

This section begins our discussion of point cloud tasks. Point cloud tasks were initially introduced by PointNet Qi et al. (2017a), which empirically explored point cloud-based tasks. Given the stronger necessity of convolutional architectures for point cloud data, it is noteworthy that current kernel regression methods rarely focus on this domain. We posit that CNTK-based kernel regression offers significant promise for point cloud applications. In the following, we provide an in-depth analysis and empirical evaluation of both CNTK and the combined CNTK-NTK kernels on point cloud regression tasks.

### 5.1 EXPERIMENTAL SETUP

It was found that encoders based on shared-MLP outperform typical MLPs. The shared-MLP in PointNet essentially corresponds to a one-dimensional convolution with a convolutional kernel size of 1 in our framework. In practice, shared-MLP typically follows a structure in which a few layers of MLP are added only after the pooling layers. Many practical convolutional networks connect MLP layers after pooling, which is precisely the purpose of introducing CNTK-NTK. For point cloud recognition, we fully adopt the PointNet architecture to propose the **PointNTK**, an instantiation of CNTK-NTK, as set up in which $L_1 = 4, L_2 = 3$ and $d_1 = 1, d_2 = 0$.

We applied the discussed kernel in the previous section to actual point cloud classification datasets. In the following experiments, we utilized three datasets from ModelNet10 and ModelNet40. The input of ModelNet10 includes both point cloud positional information and three-dimensional orientation information. We refer to the experiment using only the point coordinate information from ModelNet10 as ModelNet10$_3$, while the experiment using the complete original data of ModelNet10 is referred to as ModelNet10$_6$. Details of the experimental setup are provided in the final subsection of this section.

### 5.2 POINTNTK VERSUS POINTNET

In this section, we compare PointNTK with PointNet, where PointNet maintains all its original settings, including max pooling. Both PointNTK and PointNet are seven-layer neural networks. The first four layers are one-dimensional convolution layers with a kernel size of 1, followed by three MLP layers.

Additionally, we introduce the following comparisons:

- PointNet$_m$: A variant of PointNet with two differences: I. The training loss is replaced with the least-squares loss; II. The pooling layer is replaced with average pooling. Both networks retain their original batch normalization (BN) layers.

- 7-layer 1dCNTK and 7-layer NTK: These models serve as additional baselines for evaluation, which represents single-CNTK compared with CNTK-NTK. We set their depth to 7 in order

Table 2: Performance on ModelNet10$_3$, ModelNet10$_6$ and ModelNet40 of various structure. The superscript 'f' indicates that only 100 samples from the dataset were used as the training set. The superscript 'M' and 'L' indicate 'medium' and large with depth 9 and 15 layers respectively for 1d convolution layers.

| Datasets | ModelNet10$_3$ | ModelNet10$_3^f$ | ModelNet10$_6$ | ModelNet10$_6^f$ | ModelNet40 | ModelNet40$^f$ |
|---|---|---|---|---|---|---|
| PointNet | $91.12 \pm 1.54$ | $68.92 \pm 1.05$ | $92.08 \pm 1.75$ | $70.22 \pm 0.58$ | $88.71 \pm 0.54$ | $38.32 \pm 0.97$ |
| PointNet_m | $89.98 \pm 2.38$ | $28.22 \pm 1.56$ | $90.43 \pm 2.01$ | $27.97 \pm 1.85$ | $83.93 \pm 1.67$ | $5.17 \pm 1.22$ |
| NTK | 17.58 | 11.12 | 16.19 | 11.78 | 11.10 | 3.69 |
| 1dCNTK | 76.65 | 71.03 | 91.96 | 81.16 | 60.62 | 45.22 |
| PointNTK | 86.56 | 73.68 | 91.19 | 78.52 | 80.47 | 45.71 |
| PointNet$^M$ | $91.98 \pm 0.98$ | $69.58 \pm 0.62$ | $92.42 \pm 0.85$ | $67.33 \pm 2.65$ | $86.93 \pm 1.16$ | $30.28 \pm 1.42$ |
| PointNet$_m^M$ | $90.25 \pm 1.87$ | $13.07 \pm 2.05$ | $--$ | $--$ | $--$ | $--$ |
| PointNTK$^M$ | 87.56 | 74.23 | 92.40 | 80.04 | 81.93 | 47.77 |
| PointNet$^L$ | $91.51 \pm 0.87$ | $56.52 \pm 0.37$ | $91.53 \pm 1.89$ | $57.39 \pm 3.42$ | $85.32 \pm 1.20$ | $23.62 \pm 0.52$ |
| PointNet$_m^L$ | $88.54 \pm 2.56$ | $7.43 \pm 2.39$ | $--$ | $--$ | $--$ | $--$ |
| PointNTK$^L$ | 87.67 | 74.01 | 92.62 | 82.38 | 82.50 | 47.77 |

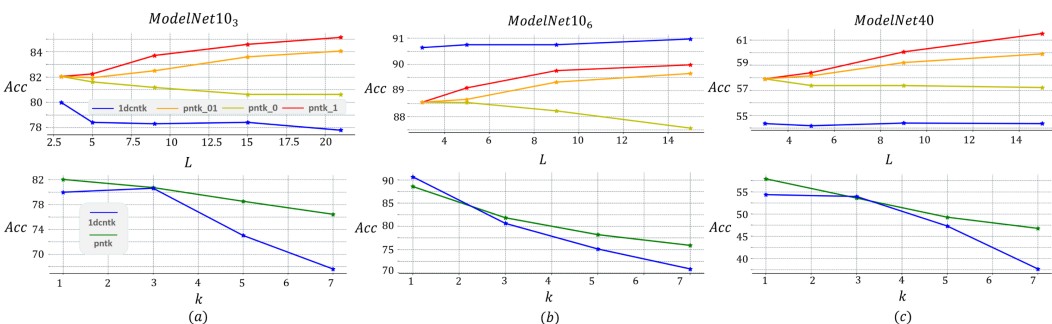

Figure 4: The ablation experiments on kernel regression based on NTK families across three datasets: the upper part shows the results for different network depths, while the lower part shows the results for different convolution kernel lengths.

to match the depth of PointNTK, which consists of 4 layers of 1D convolutional layers and 3 fully connected layers.

We conducted experiments on three datasets, where the superscript $f$ indicates that only 100 fixed samples from the dataset (covering all categories) were used for training, while the entire test set was used for evaluation. The results are summarized in Table 2, with the observations: I.PointNTK outperforms the single-layer 1dCNTK with the same number of layers; II.PointNTK performs slightly worse than PointNet and PointNet$_m$ on ModelNet10$_6$, significantly underperforms on ModelNet40 and ModelNet10$_3$, but outperforms the PointNet and PointNet$_m$ with small-training set in all datasets.

To further analyze this, we conducted experiments showing how the accuracy of kernel regression-based methods and training-based methods varies with the size of the training set, as illustrated in Figure 3.

## 5.3 ABLATION STUDY FOR KERNEL SIZE AND DEPTH

In PointNet, the defined shared-MLP is essentially a one-dimensional convolution with a kernel size of 1. This concept was also adopted in the kernel regression methods in our earlier experiments. In this section, we perform ablation experiments on the kernel size and network depth within NTK-based

kernel regression methods. We conducted experiments on three datasets, using only a fixed 500 samples from each dataset for training while testing on the entire test set.

1)Depth ablation. We conducted several experiments as follows.

- 1dCNTK: Uses only one-dimensional convolutions for the kernel with a depth of "$L + 3$". We add 3 to keep it consistent with the number of layers in PointNTK, which holds 3-layer MLPs.

- $PNTK_0$ and $PNTK_1$: $PNTK_0$ fixes the first three one-dimensional convolutional layers and varies the depth of the subsequent fully connected layers; $PNTK_1$ fixes the last three fully connected layers and varies the depth of the preceding one-dimensional convolutional layers.

- $PNTK_{01}$: Simultaneously increases the depth of both parts.

From results in the upper part of Figure 4, we observe the following: I. The design of PointNTK is highly effective. For structures using only one-dimensional convolutions, increasing depth does not yield significant improvements and may even degrade performance; II. Adding fully connected layers at the end of the one-dimensional convolutional structure improves performance across all datasets when the convolutional depth is increased; III. Fixing the depth of the one-dimensional convolutional layers while increasing the depth of the fully connected layers reduces regression performance; IV. Simultaneous depth increases in both parts are less effective than increasing the depth of the one-dimensional convolutions alone. Since increasing the depth of the 1D convolutional layers in PointNTK improves accuracy, we naturally hypothesize that the same might be true for PointNet. Finally, we further compared the results of deeper kernel regression with supervised training, as shown in Table 2.

2) Kernel size ablation. For kernel size as shown in the lower part of Figure 3, all datasets consistently indicate that a kernel size of 1 is optimal. This is intuitive because point clouds are unordered, and when the kernel size exceeds 1, the points associated with the kernel may not be meaningfully related.

## 5.4 EXPERIMENTAL DETAIL

All the experiments have been implemented using Python 3.7 and Pytorch 1.8 on one NVIDIA RTX 3090 super 24GB. In addition, neither data augmentation nor post-processing (besides global normalization) was applied, which was followed by each experiment. For kernel regression, computations were implemented using CUDA code. For supervised training methods, we adopted the parameter settings from PointNet. In small-sample experiments, we fix a subset of training set for all experiments, and all tests were conducted. We note that when the input data remains unchanged, the results of kernel regression are deterministic and unique.

## 6 CONCLUSION

In this work, we investigate the spectral differences between NTK and CNTK eigenvalues on data drawn from an arbitrary distribution. We show that CNTK consistently exhibits a broader eigenvalue distribution and, on average, smaller eigenvalues compared to NTK. This provides theoretical evidence for the superior generalization ability of CNTK over NTK. Building on this insight, we define a data-dependent suitability measure based on the spectral range of the first-layer CNTK and NTK, which depends solely on the covariance matrix of the data. Our experiments reveal that point cloud tasks benefit more significantly from convolutional structures. Motivated by this, we conduct extensive kernel regression experiments on point cloud datasets and demonstrate that our proposed strategy, which combines CNTK and NTK, is particularly effective in this setting, filling an existing gap in CNTK performance on point cloud tasks.

There are still some limitations in our work. While our main theoretical results are derived without assuming any specific data distribution, computing $\beta_K$ for an arbitrary $L$-layer NTK does require distributional assumptions. As a result, in practical applications, we rely on the first layer to evaluate data suitability. However, the computation of spectral properties of $L$-layer NTK and $L$-layer CNTK is hard. Future investigations into its upper and lower bounds may yield more precise spectral characteristics of the data at layer $L$.

## 7 ACKNOWLEDGEMENT

This work was supported in part by the Jing-Jin-Ji Regional Integrated Environmental Improvement-National Science and Technology Major Project (2025ZD1208203), the National Natural Science Foundation of China (62332013), and the Collaborative Innovation Center of Novel Software Technology and Industrialization.

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

## A    USE OF LLMs

We only use the large model to modify grammatical errors and polish the articles; the large model has not been involved in any theorem proofs or the discovery of ideas.

## B    RELATED WORK

### B.1    NEURAL TANGENT KERNEL

The Neural Tangent Kernel (NTK), a widely used gradient kernel, is first introduced by Jacot et al. Jacot et al. (2018). It is formulated as a Gram matrix constructed from gradients and has been shown to be equivalent to training a fully-connected neural network under specific parameterization, resembling a kernel method, as demonstrated in Lee et al. (2020) Chen et al. (2021b). Over time, the applicability of NTK has been extended to various neural network architectures, including convolutional networks Samarin et al. (2020) Arora et al. (2019a) Li et al. (2019), residual networks Yang & Schoenholz (2017), attention-based networks Hron et al. (2020), GAN architectures Franceschi et al. (2022), and even graph neural networks Du et al. (2019b). Given its ability to capture critical properties of network architectures and datasets, NTK has also been integrated into Neural Architecture Search (NAS) Chen et al. (2021a), helping to reduce time overhead in NAS processes Xu et al. (2021) Mok et al. (2022). Moreover, NTK has proven useful as a theoretical tool for analyzing input encoding in Coordinate-MLP Tancik et al. (2020) and convergence properties of PINN Wang et al. (2022), and for explaining the effectiveness of adversarial training strategies Loo et al. (2022), active learning strategies Mohamadi et al. (2022), and ensemble learning He et al. (2020). NTK has also been used to demonstrate zero training loss in polynomial time for over-parameterized residual networks Du et al. (2019a), further underscoring its significance.

To address the computational complexity of the NTK, study Arora et al. (2019a) simplified this computation. Additionally, during real-world network training, the network width is finite. In finite-width networks, the kernel can change, and some studies Shan & Bordelon (2021) Seleznova et al. (2024) have qualitatively pointed out the pattern of this variation. The computation of the Neural Tangent Kernel in finite-width networks is more complex, and this is also discussed Dyer & Gur-Ari (2019), with implementation details available in Engel et al. (2022). These kernels play a crucial role in convergence analysis, particularly in evaluating convergence rates, depending on the minimum eigenvalue of the NTK Chen et al. (2021a). The range of NTK applications has expanded considerably since its inception. Its constant kernel property allows it to be used to estimate the performance of specific architectures by assessing convergence behavior. Notably, NTK, as a kernel, integrates seamlessly with most kernel methods Arora et al. (2019b).

### B.2    POINT CLOUD CLASSIFICATION

Extensive research has been conducted on point cloud analysis, with a primary focus on learning data-driven representations in an end-to-end fashion. The inherent flexibility of the point cloud

format has led to the emergence of innovative approaches aimed at representing 3D structures both effectively and informatively. The seminal work PointNet Qi et al. (2017a) used shared-MLP on point cloud data demonstrating that neural networks can directly process global coordinates to extract geometric information. PointNet has since become a cornerstone for numerous point cloud-based tasks. Building on this foundation, PointNet++ Qi et al. (2017b) introduced local aggregation mechanisms, significantly enhancing the ability to capture fine-grained geometric features. To further improve the extraction of local structures, methods such as KPConv Thomas et al. (2019) introduced convolutional operations to 3D data, achieving notable performance gains across diverse benchmarks. More recently, inspired by the Transformer architecture Vaswani et al. (2017), models like PCT Guo et al. (2021) effectively incorporate self-attention to better capture point cloud features, achieving state-of-the-art results in various tasks.

## C PROOF OF MAIN RESULTS

### C.1 UNIFIED CONVOLUTIONAL NEURAL NETWORK

The general $d$-dimension forward propagation CNNs are defined as follows.

**Definition 2.** *For $d$-dim convolutional neural network, given lengths of tensor $\{h_i\}_{i\in[d]}$, lengths of convolutional kernel $\{s_i\}_{i\in[d]}$ and widths of neural network $\{n_0, n_1, ..., n_L\}$ with $L$-layer. The $d$-dim convolutional neural network (without head) is defined as $\boldsymbol{f} : \mathbb{R}^{\Pi_{i=1}^d h_i \times n_0} \to \mathbb{R}^{\Pi_{i=1}^d h_i \times n_{L-1}}$ with forward propagation:*

$$
\begin{aligned}
\tilde{\boldsymbol{f}}^{(l)}(x) &:= \phi(\boldsymbol{f}^{(l)}(x)), x^0(x) := x. \\
\boldsymbol{f}^{(l)}(x) &:= \boldsymbol{W}^{(l)} * \tilde{\boldsymbol{f}}^{(l-1)}(x) + \boldsymbol{b}^{(l)}, \quad 1 \le l \le L-1 \\
\boldsymbol{f}(x) &:= \tilde{\boldsymbol{f}}^{L-1}(x).
\end{aligned}
\tag{B-1}
$$

Input $x$ is a $d$-tensor from $\mathbb{R}^{\Pi_{i=1}^d h_i}$, $\mathbb{R}^{\Pi_{i=1}^d h_i}$ denotes space $\mathbb{R}^{h_1 \times h_2 \times ... \times h_d}$. The activation function $\phi(x)$ is Relu in this paper, The so-called "standard initialization" denotes that the defined convolution kernel $\mathcal{W}_{ij}^{(l)} \in \mathbb{R}^{\Pi_{i=1}^d s_i}$ and bias $b^{(l)} \in \mathbb{R}^{\times_{i=1}^d h_i}$. All entries of any $\mathcal{W}_{ij}^{(l)}$ are from $\mathcal{N}\left(0, \frac{\sigma_w^2}{n_{l-1} \times \Pi_{i=1}^d s_i}\right)$ independently, and all entries of any $b_j^{(l)}$ are from $\mathcal{N}\left(0, \sigma_b^2\right)$ independently. $\boldsymbol{W}^{(l)}$ is still called 'matrix' that is defined in $\left(\mathbb{R}^{\Pi_{i=1}^d s_i}\right)^{n_{l-1} \times n_l}$; $\boldsymbol{b}^{(l)}$ is still called 'vector' defined in $\left(\mathbb{R}^{\times_{i=1}^d h_i}\right)^{n_l}$. Convolution operator $* : \mathbb{R}^{\Pi_{i=1}^d h_i} \times \mathbb{R}^{\Pi_{i=1}^d s_i} \to \mathbb{R}^{\Pi_{i=1}^d h_i}$ is defined as:

$$
\begin{aligned}
[w * f][i_1, i_2, ..., i_d] &= \sum_{j_i \in [1, s_i]} w[j_1, j_2, ..., j_d] \\
&\times f[j_1 + i_1 - s_1/2, j_2 + i_2 - s_2/2, ..., j_d + i_d - s_d/2].
\end{aligned}
\tag{B-2}
$$

The operator $*$ for matrix still satisfies matrix multiplication. And we suppose that:

$$
\exists k \ s.t. \ i_k \le 0 \Rightarrow f[i_1, i_2, ..., i_d] = 0,
\tag{B-3}
$$

which is zero-padding.

In neural network applications, the corresponding head is typically added based on the specific task. This paper discusses classification network. We define two types of heads, one of which refers to the pooling layer and the other is naive.

**Definition 3.** *Define pooling function $\mathcal{P} : \mathbb{R}^{\times_{i=1}^d h_i \times n} \to \mathbb{R}^n$. $d$-dim convolutional neural network for classification task, without pooling in the last layer $\boldsymbol{f}_{wop}$, is defined as:*

$$
\begin{aligned}
\boldsymbol{f}_{wop}(x) &= \mathcal{P}_{av}\left(\boldsymbol{W}^{(L)} * \boldsymbol{f}(x) + \boldsymbol{b}^{(L)}\right), \\
\boldsymbol{W}^{(L)} &\in \left(\mathbb{R}^{\times_{i=1}^d s_i}\right)^{n_{L-1} \times n_L}, \boldsymbol{b}^{(L)} \in \left(\mathbb{R}^{\times_{i=1}^d h_i}\right)^{n_L},
\end{aligned}
\tag{B-4}
$$

*with pooling $\boldsymbol{f}_{wp}$ is defined as:*

$$
\begin{aligned}
\boldsymbol{f}_{wp}(x) &= \boldsymbol{W}^{(L)} * \mathcal{P}\left(\boldsymbol{f}(x)\right) + \boldsymbol{b}^{(L)}, \\
\boldsymbol{W}^{(L)} &\in \mathbb{R}^{n_{L-1} \times n_L}, \boldsymbol{b}^{(L)} \in \mathbb{R}^{n_L}.
\end{aligned}
\tag{B-5}
$$

## C.2 Lemma 1: CNTK Formula

We first consider function $\boldsymbol{f} : \mathbb{R}^{\times_{i=1}^{d} h_i \times n_0} \to \mathbb{R}^{\times_{i=1}^{d} h_i \times n_{L-1}}$ without head, as in practice, we suppose that $s_i = s$ for any $i$. For entries $f_p^{(l)}(1 \le l \le L-1$, and note this symbol omits a subscript $j$ that denotes the $f_p^{(l)} = \boldsymbol{f}_{j,p}^{(l)}$. As can be seen in the paper, this notation can actually be omitted. in which $p = \{p_1, p_2, ..., p_d\}$. Furthermore, we define neighborhood of $p$ as $N(p) = \{\cap_{i=1}^{d}(p_i - s/2, p_i + s/2) \cap \mathbb{Z} | 1 \le i \le d\}$. Kernel of samples $x, x'$ as entry is induced by:

$$
\begin{aligned}
K_{p,q}^{(l)}(x, x') &= \nabla_\theta^T f_p^{(l)}(x) \nabla_\theta f_q^{(l)}(x') \\
&= \nabla_\theta^T \sum_{i=1}^{n_l} \left( \mathcal{W}_{ij}^{(l)} * \phi_{Np}(f_i^{(l-1)}(x)) + b_{j,N(p)}^{(l)} \right) \nabla_\theta \sum_{i=1}^{n_l} \left( \mathcal{W}_{ij}^{(l+1)} * \phi_{Nq}(f_i^{(l-1)}(x')) + b_{j,N(q)}^{(l)} \right) \\
&= \nabla_\theta^T \sum_{i=1}^{n_l} \sum_{p' \in N(p)} \mathcal{W}_{ij,p'-p}^{(l)} \times \phi_{p'}(f_i^{(l-1)}(x)) \nabla_\theta \sum_{i=1}^{n_l} \sum_{q' \in N(q)} \mathcal{W}_{ij,q'-q}^{(l)} \times \phi_{q'}(f_i^{(l-1)}(x')) + \sigma_b^2 \\
&= \frac{1}{n_l} \sum_{i=1}^{n_l} (w_{ij}^{(l+1)})^2 \sum_{p' \in N(p)} \sum_{q' \in N(q)} \nabla_\theta^T f_{i,p'}^{(l-1)}(x) \nabla_\theta f_{i,q'}^{(l-1)}(x') \dot{\phi}_{p'}(f_i^{(l-1)}(x)) \dot{\phi}_{q'}(f_i^{(l-1)}(x')) \\
&\quad + \frac{1}{n_l} \sum_{i=1}^{n_l} \sum_{p' \in N(p)} \sum_{q' \in N(q)} \phi_{p'}(f_i^{(l-1)}(x)) \phi_{q'}(f_i^{(l-1)}(x')) + \sigma_b^2
\end{aligned}
\tag{B-6}
$$

In the calculation process of the tangent kernel, the presence of the bias term introduces a shift in each recursion step. Therefore, in many tangent kernel computations, the bias term is excluded. This article follows the same approach that $\sigma_b = 0$. For convenience, we need some symbols to record intermediate quantities. For $l = 0$:

$$
\begin{aligned}
\mathbf{\Sigma}^{(0)}(x, x') &= \sum_{i=0}^{n_0} x_i \otimes x_i' \\
\Sigma_{p,q}^{(0)}(x, x') &= tr\left( \mathbf{\Sigma}_{N(p),N(q)}^{(0)}(x, x') \right).
\end{aligned}
\tag{B-7}
$$

in which $\otimes$ denotes tensor product. For $1 \le l \le L-1$, define the covariance matrix, the 0th-order gradient term, and the 1st-order gradient term in sequence according to the definition order in NTK.

$$
\begin{aligned}
covariance\ matrix : \mathbf{\Lambda}_{p,q}^{(l)}(x, x') &= \begin{pmatrix} \Sigma_{p,q}^{(l-1)}(x, x) & \Sigma_{p,q}^{(l-1)}(x, x') \\ \Sigma_{p,q}^{(l-1)}(x', x) & \Sigma_{p,q}^{(l-1)}(x', x') \end{pmatrix} \in \mathbb{R}^{2 \times 2}. \\
0th - order\ term : \qquad \mathbf{\Sigma}_{p,q}^{*(l)}(x, x') &= \mathbb{E}_{(u,v) \sim \mathcal{N}\left(\mathbf{0}, \mathbf{\Lambda}_{p,q}^{(l)}(x,x')\right)} [\phi(u)\phi(v)]. \\
1th - order\ term : \qquad \dot{\mathbf{\Sigma}}_{p,q}^{*(l)}(x, x') &= \mathbb{E}_{(u,v) \sim \mathcal{N}\left(\mathbf{0}, \mathbf{\Lambda}_{p,q}^{(l)}(x,x')\right)} \left[ \dot{\phi}(u)\dot{\phi}(v) \right].
\end{aligned}
\tag{B-8}
$$

And the induction:

$$
\mathcal{E}_{p,q}^{(l)}(x, x') = tr\left( \mathbf{\Sigma}_{Np),N(q)}^{*(l)}(x, x') \right).
\tag{B-9}
$$

Now we can present the recursive formula for the kernel as follows:

$$
\begin{aligned}
K_{p,q}^{(l)}(x, x') &= \frac{1}{n_l} \sum_{i=1}^{n_l} (w_{ij}^{(l+1)})^2 \sum_{p' \in N(p)} \sum_{q' \in N(q)} \nabla_\theta^T f_{i,p'}^{(l-1)}(x) \nabla_\theta f_{i,q'}^{(l-1)}(x') \dot{\phi}_{p'}(f_i^{(l-1)}(x)) \dot{\phi}_{q'}(f_i^{(l-1)}(x')) \\
&\quad + \frac{1}{n_l} \sum_{i=1}^{n_l} \sum_{p' \in N(p)} \sum_{q' \in N(q)} \phi_{p'}(f_i^{(l-1)}(x)) \phi_{q'}(f_i^{(l-1)}(x')) + \sigma_b^2 \\
&= K_{N(p),N(q)}^{(l-1)} * \dot{\mathbf{\Sigma}}_{N(p),N(q)}^{(l)}(x, x') + \mathcal{E}_{p,q}^{(l)}(x, x').
\end{aligned}
\tag{B-10}
$$

Now, we consider a neural network $f_{wop}$, $f_{wp}$ with a head (Assume that the final layer has only one output; the case with multiple outputs is analogous) as follows:

$$
\begin{aligned}
K_{wop}(x, x') &= \nabla_\theta^T f_{wop}(x) \nabla_\theta f_{wop}(x') \\
&= \nabla_\theta^T \mathcal{P}_{av} \left( \sum_{i=1}^{n_{L-1}} \mathcal{W}_i^{(L)} * \boldsymbol{f}(x) \right) \nabla_\theta \mathcal{P}_{av} \left( \sum_{i=1}^{n_{L-1}} \mathcal{W}_i^{(L)} * \boldsymbol{f}(x') \right) \\
&= \frac{1}{(\Pi_{i=1}^d h_i)^2} \sum_{p \in [h_1, h_2, \ldots, h_d]} K_{p,p}^{(L)}(x, x') \\
&= tr \left( K^{(L)}(x, x') \right)
\end{aligned}
\tag{B-11}
$$

$$
\begin{aligned}
K_{wp}(x, x') &= \nabla_\theta^T f_{wp}(x) \nabla_\theta f_{wp}(x') \\
&= \nabla_\theta^T \sum_{i=1}^{n_{L-1}} \mathcal{W}_i^{(L)} \times \mathcal{P}_{av} (\boldsymbol{f}(x)) \nabla_\theta \sum_{i=1}^{n_{L-1}} \mathcal{W}_i^{(L)} \times \mathcal{P}_{av} (\boldsymbol{f}(x')) \\
&= \frac{1}{n_{L-1}} \sum_{i=1}^{n_{L-1}} \mathcal{P}_{av} (\boldsymbol{f}(x)) \times \mathcal{P}_{av} (\boldsymbol{f}(x')) + \\
&\quad \frac{1}{n_{L-1}} \sum_{i=1}^{n_{L-1}} (w_i^{(L)})^2 \left( \frac{1}{\Pi_{i=1}^d h_i} \sum_{p \in [h_1, h_2, \ldots, h_d]} \sum_{q \in [h_1, h_2, \ldots, h_d]} \nabla_\theta^T \boldsymbol{f}_{i,p}(x) \nabla_\theta \boldsymbol{f}_{i,q}(x') \right) \\
&= \mathcal{P}_{av} \left( K^{(L)}(x, x') \right) + \mathcal{P}_{av} \left( \boldsymbol{\Sigma}_{p,q}^{(L)}(x, x') \right)
\end{aligned}
\tag{B-12}
$$

If we fix the last layer, the upper eqaution can be redued to our proposition:

$$
K_{wp}(x, x') = \mathcal{P}_{av} \left( K^{(L)}(x, x') \right)
\tag{B-13}
$$

### C.3 PROPOSITION 1: NTK FOLLOWED BY CNTK

The calculation of the composite-NTK only requires knowledge of the propagation of NTK at the pooling layer, $K_{wp}$, and the propagation of covariance, which necessitates computing $Cov(f(x), f(x'))$:

$$
\begin{aligned}
&Cov \left( P_{av}^{d_1 \to d_2} (\boldsymbol{f}(x)), P_{av}^{d_1 \to d_2} (\boldsymbol{f}(x)) \right) \\
&= \frac{1}{\Pi_{i=d_2+1}^{d_1} h_i^1} \sum_{i_{d_2+1}, i_{d_2+2}, \ldots, i_{d_1}} Cov \left( \boldsymbol{f}_{\ldots, i_{d_2+1}, i_{d_2+2}, \ldots, i_{d_1}}(x), \boldsymbol{f}_{\ldots, i_{d_2+1}, i_{d_2+2}, \ldots, i_{d_1}}(x') \right) \\
&= \mathcal{P}_{av}^{d_1 \to d_2} (\Sigma(x, x'))
\end{aligned}
\tag{B-14}
$$

When $d_2 = 0$, it corresponds to the common average pooling layer, and the covariance tensor is:

$$
\Sigma^{(L+1)}(x, x') = \mathcal{P}_{av} \left( \boldsymbol{\Sigma}^{(L)}(x, x') \right)
\tag{B-15}
$$

## D SPECTRAL COMPARISON

**Definition 4.** *Suppose tensor $x \in \mathbb{R}^{\times_{i=1}^d h_i \times n_0}$, we mark $x_{\{i,p\}}^{(j)}$ as the $\{i, p\} - entries$ of the $j - th$ samples $x^{(j)}$ in which $i \in [n_0]$ and $p \in [[h_i]_d]$. We further define scalar $H = \Pi_{i=1}^d h_i$. For any matrix $K$, let $m_K = \frac{1}{N} tr(K)$ and $s_K = \frac{1}{N^2} tr(KK^\top)$*

**Assumption 1.** *We assume that tasks with random tensor $x_{k,.} \in \mathbb{R}^{\times_{i=1}^d h_i}$, in which all-entries of $x_{k,.}$ are sampled from a distribution with covariance matrix $\boldsymbol{\sigma}^{(k)} \in \mathbb{R}^{H \times H}$, the scalar $\sigma_{p,q}^{(k)}$ denotes covariance of random variable $x_{k,p}$ and $x_{k,q}$. And for any $k \neq k'$, any entries of $x_{\{k,.\}}$ and $x_{\{k',.\}}$ are independent. Furthermore, we assume any sample $x^{(i)}$ of datasets $\{x^{(i)}\}_{i \in [N]}$ is sampled from the above distribution independently.*

**Lemma 3.** *Given a $N \times N$ positive definite matrix $K$, let its set of eigenvalues be denoted as $\lambda_1 \leq \lambda_2 \leq ... \leq \lambda_N$, then:*

$$m_K = \frac{1}{N} \sum_{i=1}^{N} \lambda_i, \quad s_K = \frac{1}{N^2} \sum_{i=1}^{N} \lambda_i^2, \tag{B-16}$$

*Proof.* For a $N \times N$ positive definite matrix $K$, by researchHorn & Johnson (2012) Cai et al. (2024):

$$\mu = \frac{1}{N} \sum_{i=1}^{N} \lambda_i = \frac{1}{N} tr(K) = m_K \tag{B-17}$$

$$\sigma^2 = \frac{1}{n} \sum_{i=1}^{N} (\lambda_i - \mu)^2 = \frac{1}{N} tr(KK^\top) - \mu^2 = Ns_K - m_K^2 \tag{B-18}$$

let $\lambda_{min} = \lambda_1, \lambda_{max} = \lambda_N$, then:

$$\lambda_{max} \geq \frac{\sum_{i=1}^{N} \lambda_i^2}{\sum_{i=1}^{N} \lambda_i} = \frac{Ns_K}{m_K}, \quad \lambda_{min} \leq \frac{1}{N} \sum_{i=1}^{N} \lambda_i = m_K \tag{B-19}$$

therefore:

$$\frac{\lambda_{max}}{\lambda_{min}} \geq N\beta_K. \tag{B-20}$$

$\square$

## D.1 Proof of Theorems in Content

**Lemma 4.** *For a set of tensors(datasets) $\{x^{(i)}\}$. For the $L - layer$ NTK and CNTK. The following satisfies:*

$$\Sigma^{(l)}(x,x) \geq \frac{1}{H^2} \sum_{p,q \in [[h_i]_d]} \mathbf{\Sigma}_{p,q}^{(l)}(x,x), \quad \dot{\Sigma}^{(l)}(x,x) \geq \dot{\mathbf{\Sigma}}_{p,q}^{(l)}(x,x). \tag{B-21}$$

*Proof.* I. when $l = 0$, as:

$$\Sigma^{(0)}(x,x) - \frac{1}{H^2} \mathbf{\Sigma}_{p,q}^{(0)}(x,x) = \frac{1}{n_0} \sum_{i=1}^{n_0} \left( \frac{1}{H} \sum_{p \in [[h_i]_d]} x_{i,p}^2 - \frac{1}{H^2} \sum_{p,q \in [[h_i]_d]} x_{i,p}x_{i,q} \right) \geq 0 \tag{B-22}$$

II. suppose that $\Sigma^{(l-1)}(x,x) \geq \frac{1}{H^2} \sum_{p,q \in [[h_i]_d]} \mathbf{\Sigma}_{p,q}^{(l-1)}(x,x)$ satisfies for $l \geq 1$, then:

$$
\begin{aligned}
\Sigma^{(l)}(x,x) = \kappa(1, \Sigma^{(l-1)}(x,x)) &\geq \kappa(1, \frac{1}{H^2} \sum_{p,q \in [[h_i]_d]} \mathbf{\Sigma}_{p,q}^{(l)}(x,x)) \\
&\geq \frac{1}{H^2} \sum_{p,q \in [[h_i]_d]} \kappa(1, \mathbf{\Sigma}_{p,q}^{(l)}(x,x)) \\
&\geq \frac{1}{H^2} \sum_{p,q \in [[h_i]_d]} \kappa\left( \frac{\mathbf{\Sigma}_{p,q}^{(l-1)}(x,x)}{\sqrt{\mathbf{\Sigma}_{p,p}^{(l-1)}(x,x)\mathbf{\Sigma}_{q,q}^{(l-1)}(x,x)}}, \mathbf{\Sigma}_{p,q}^{(l)}(x,x) \right) \\
&= \frac{1}{H^2} \sum_{p,q \in [[h_i]_d]} \mathbf{\Sigma}_{p,q}^{(l-1)}(x,x)
\end{aligned}
\tag{B-23}
$$

$\dot{\Sigma}^{(l)}(x,x) \geq \dot{\mathbf{\Sigma}}_{p,q}^{(l)}(x,x)$ is similar as:

$$\dot{\Sigma}^{(l)}(x,x) = \dot{\kappa}(1) \geq \dot{\mathbf{\Sigma}}_{p,q}^{(l)}(x,x) \tag{B-24}$$

$\square$

**Assumption 2.** *When $\sigma^{(i)}_{\{p,p\}} = \sigma^{(j)}_{\{q,q\}}$, we define the dual-activation function and one-order dual-activation function as $\kappa(\lambda, c)$ which is concave about $c$ and increasing about both $\lambda, c$, that is for any $\lambda, c_1, c_2, a \in [0, 1]$:*

$$\kappa(\lambda, ac_1 + (1-a)c_2) \geq a\kappa(\lambda, c_1) + (1-a)\kappa(\lambda, c_2). \tag{B-25}$$

*$\dot{\kappa}(\lambda)$ is increasing about $\lambda$ and independent to $c_1 c_2$.*

Notice that for the Relu function it satisfies, as it is linear w.r.t. item $c$.

**Theorem 5.** *For a set of tensors(datasets) $\{x^{(i)}\}$. Let $m^{(L)}_{CNTK}, m^{(L)}_{NTK}$ be denoted as the $m_K$ of L-layer CNTK and NTK. The following satisfies:*

$$m^{(L)}_{CNTK} \leq m^{(L)}_{NTK}. \tag{B-26}$$

*Proof.* It satisfies in $L = 0$, suppose it satisfies in $L - 1$, by Lemma 4, the following holds:

$$
\begin{aligned}
m^{(L)}_{CNTK} = K^{(L)}_{CNTK}(x, x) &= \mathcal{P}_{av}\left( \dot{\mathbf{\Sigma}}^{(L)}(x, x) \odot \mathbf{K}^{(L-1)}_{CNTK}(x, x) + \mathbf{\Sigma}^{(L)}(x, x) \right) \\
&\leq \dot{\Sigma}^{(L)}(x, x)\mathcal{P}_{av}\left( \mathbf{K}^{(L-1)}_{CNTK}(x, x) \right) + \mathcal{P}_{av}\left( \mathbf{\Sigma}^{(L)}(x, x) \right) \\
&\leq \dot{\Sigma}^{(L)}(x, x)K^{(L-1)}_{NTK}(x, x) + \Sigma^{(L)}(x, x) = m^{(L)}_{NTK}.
\end{aligned}
\tag{B-27}
$$

$\square$

**Proposition 6.** *For a set of tensors (datasets) $\{x^{(i)}\}_{i \in [N]}$. Let $N \to \infty$. When $K = K^{(0)}_{NTK}$, then:*

$$
\begin{aligned}
m_K &= \frac{1}{n_0}\sum_{j=1}^{n_0} mean(diag(\boldsymbol{\sigma}^{(j)})) \\
s_K &= \frac{1}{n_0^2}\sum_{j,j' \in [n_0]} mean(\boldsymbol{\sigma}^{(j)} \odot \boldsymbol{\sigma}^{(j')}),
\end{aligned}
\tag{B-28}
$$

*when $K = K^{(0)}_{CNTK}$, then:*

$$
\begin{aligned}
m_K &= \frac{1}{n_0}\sum_{j=1}^{n_0} mean(\boldsymbol{\sigma}^{(j)}) \\
s_K &= \frac{1}{n_0^2}\sum_{j,j' \in [n_0]} mean(\boldsymbol{\sigma}^{(j)} \otimes \boldsymbol{\sigma}^{(j')}),
\end{aligned}
\tag{B-29}
$$

*in which $\odot$ denotes Schur product (element-wise product), $\otimes$ denotes Kronecker product (tensor prodcut).*

*Proof.* When $K = K^{(0)}_{NTK}$:

$$
\begin{aligned}
m_K &= \frac{1}{N}tr(K) = \frac{1}{N}\sum_{i=1}^{N} K(x^{(i)}, x^{(i)}) \\
&= \frac{1}{n_0 N H}\sum_{i=1}^{N}\sum_{j=1}^{n_0}\sum_{p \in [[h_k]_d]} (x^{(i)}_{\{j,p\}})^2 \\
&\stackrel{N \to \infty}{\Longrightarrow} \frac{1}{n_0 H}\sum_{j=1}^{n_0}\sum_{p \in [[h_k]_d]} \sigma^{(j)}_{\{p,p\}} \\
&= \frac{1}{n_0}\sum_{j=1}^{n_0} mean(diag(\boldsymbol{\sigma}^{(j)}))
\end{aligned}
\tag{B-30}
$$

$$s_K = \frac{1}{N^2} tr(KK\top) = \frac{1}{N^2} \sum_{i,j \in [N]} K^2(x^{(i)}, x^{(j)})$$

$$= \frac{1}{n_0^2 N^2 H^2} \sum_{i,i' \in [N]} (\sum_{j=1}^{n_0} \sum_{p \in [[h_k]_d]} x_{\{j,p\}}^{(i)} x_{\{j,p\}}^{(i')})^2$$

$$= \frac{1}{n_0^2 N^2 H^2} \sum_{i=i' \in [N]} \sum_{j,j' \in [n_0]}^{n_0} \sum_{p,q \in [[h_k]_d]} x_{\{j,p\}}^{(i)} x_{\{j',p\}}^{(i')} x_{\{j,q\}}^{(i)} x_{\{j',q\}}^{(i')}$$

$$+ \frac{1}{n_0^2 N^2 H^2} \sum_{i \neq i' \in [N]} \sum_{j,j' \in [n_0]}^{n_0} \sum_{p,q \in [[h_k]_d]} x_{\{j,p\}}^{(i)} x_{\{j',p\}}^{(i')} x_{\{j,q\}}^{(i)} x_{\{j',q\}}^{(i')} \qquad \text{(B-31)}$$

$$\overset{N \to \infty}{\Longrightarrow} O(\frac{1}{N}) + \frac{1}{n_0^2 H^2} \mathbb{E} \left( \sum_{j,j' \in [n_0]}^{n_0} \sum_{p,q \in [[h_k]_d]} x_{\{j,p\}}^{(i)} x_{\{j',p\}}^{(i')} x_{\{j,q\}}^{(i)} x_{\{j',q\}}^{(i')} \right)$$

$$= \frac{1}{n_0^2 H^2} \sum_{j,j' \in [n_0]}^{n_0} \sum_{p,q \in [[h_k]_d]} \sigma_{\{p,q\}}^{(j)} \sigma_{\{p,q\}}^{(j')}$$

$$= \frac{1}{n_0^2} \sum_{j,j' \in [n_0]}^{n_0} mean(\boldsymbol{\sigma}^{(j)} \odot \boldsymbol{\sigma}^{(j')})$$

when $K = K_{CNTK}^{(0)}$:

$$m_K = \frac{1}{N} tr(K) = \frac{1}{N} \sum_{i=1}^{N} K(x^{(i)}, x^{(i)})$$

$$= \frac{1}{n_0 N H^2} \sum_{i=1}^{N} \sum_{j=1}^{n_0} \sum_{p,q \in [[h_k]_d]} x_{\{j,p\}}^{(i)} x_{\{j,q\}}^{(i)}$$

$$\overset{N \to \infty}{\Longrightarrow} \frac{1}{n_0 H^2} \sum_{j=1}^{n_0} \sum_{p,q \in [[h_k]_d]} \sigma_{\{p,q\}}^{(j)}, \qquad \text{(B-32)}$$

$$= \frac{1}{n_0} \sum_{j=1}^{n_0} mean(\boldsymbol{\sigma}^{(j)})$$

$$s_K = \frac{1}{N^2} tr(KK\top) = \frac{1}{N^2} \sum_{i,j \in [N]} K^2(x^{(i)}, x^{(j)})$$

$$= \frac{1}{n_0^2 N^2 H^4} \sum_{i,i' \in [N]} (\sum_{j=1}^{n_0} \sum_{p,q \in [[h_k]_d]} x_{\{j,p\}}^{(i)} x_{\{j,q\}}^{(i')})^2$$

$$= \frac{1}{n_0^2 N^2 H^4} \sum_{i=i' \in [N]} \sum_{j,j' \in [n_0]} \sum_{p,q,p',q' \in [[h_k]_d]} x_{\{j,p\}}^{(i)} x_{\{j',q\}}^{(i')} x_{\{j,p'\}}^{(i)} x_{\{j',q'\}}^{(i')}$$

$$+ \frac{1}{n_0^2 N^2 H^4} \sum_{i \neq i' \in [N]} \sum_{j,j' \in [n_0]} \sum_{p,q,p',q' \in [[h_k]_d]} x_{\{j,p\}}^{(i)} x_{\{j',q\}}^{(i')} x_{\{j,p'\}}^{(i)} x_{\{j',q'\}}^{(i')} \qquad \text{(B-33)}$$

$$\overset{N \to \infty}{\Longrightarrow} O(\frac{1}{N}) + \frac{1}{n_0^2 H^4} \mathbb{E} \left( \sum_{j,j' \in [n_0]} \sum_{p,q \in [[h_k]_d]} x_{\{j,p\}}^{(i)} x_{\{j',q\}}^{(i')} x_{\{j,p'\}}^{(i)} x_{\{j',q'\}}^{(i')} \right)$$

$$= \frac{1}{n_0^2 H^4} \sum_{j,j' \in [n_0]} \sum_{p,q,p',q' \in [[h_k]_d]} \sigma_{\{p,q\}}^{(j)} \sigma_{\{p',q'\}}^{(j')}$$

$$= \frac{1}{n_0^2} \sum_{j,j' \in [n_0]} mean(\boldsymbol{\sigma}^{(j)} \otimes \boldsymbol{\sigma}^{(j')})$$

$\square$

**Lemma 5.** *For a set of tensors (datasets) $\{x^{(i)}\}_{i\in[N]}$. Let $N \to \infty$, define the **standard spectral bandwidth** $\beta_K = \frac{s_K}{m_K^2}$, by research Cai et al. (2024):*

$$\beta_K = \frac{\sum_{i=1}^{N} \lambda_i^2}{\left(\sum_{i=1}^{N} \lambda_i\right)^2}$$

*For the foundation:*

$$\beta_{CNTK^{(0)}} = 1$$
$$\beta_{NTK^{(0)}} = \frac{\sum_{j,j'\in[n_0]} mean(\boldsymbol{\sigma}^{(j)} \odot \boldsymbol{\sigma}^{(j')})}{\sum_{j,j'\in[n_0]} mean(diag(\boldsymbol{\sigma}^{(j)}) \otimes diag(\boldsymbol{\sigma}^{(j')}))} \le 1 \tag{B-34}$$

When we further assume that the covariance matrix is the same across all channels, the conclusion stated in the main text follows.

**Theorem 7.** *For a set of tensors(datasets) $\{x^{(i)}\}$. Let $N \to \infty$, define the **standard spectral bandwidth** $\beta_K = \frac{s_K}{m_K^2}$, for the $L$-layer NTK and $L'$-layer CNTK. The following satisfies;*

$$1 = \beta_{CNTK}^{(L')} \ge \beta_{NTK}^{(L)} \tag{B-35}$$

*Proof.* For this theorem, we obtain a rather strong result; in fact, it suffices to prove statements $\beta_{CNTK}^{(L')} = 1$ and $\beta_{NTK}^{(L)} \le 1$ separately. Recall the proof in Proposition 7. It suffices to replace $x$ with the random variable appearing before the final linear layer in CNTK (or NTK) to complete the proof, since the above argument relies neither on the specific data distribution nor on a particular instance of the covariance matrix. $\square$

### D.2 UPPER BOUND OF $\Sigma$ IN NTK AND CNTK WITH RELU

**Lemma 6.** *Suppose that the activation function $\phi$ is $\sqrt{2}Relu$. Then in NTK:*

$$\Sigma^{(l)}(x,x) = K^{(0)}(x,x), \ \dot{\Sigma}^{(l)}(x,x) = 1. \tag{B-36}$$

*Proof.* For the step $l$, $\Sigma^{(l)}(x,x)$ follows a recursion:

$$\Sigma^{(l)}(x,x) = \frac{\lambda(\pi - arccos(\lambda)) + \sqrt{1-\lambda^2}}{\pi} \cdot c_1 c_2, \ \dot{\Sigma}^{(l)}(x,x) = \frac{\pi - arccos(\lambda)}{\pi}. \tag{B-37}$$

In which $\boldsymbol{D\Lambda D} = \begin{pmatrix} \Sigma^{(l-1)}(x,x) & \Sigma^{(l-1)}(x,x) \\ \Sigma^{(l-1)}(x,x) & \Sigma^{(l-1)}(x,x) \end{pmatrix}$, thus $c_1 c_2 = \Sigma^{(l-1)}(x,x)$ and $\lambda = 1$. Therefore:

$$\Sigma^{(l)}(x,x) = \Sigma^{(l-1)}(x,x), \ \dot{\Sigma}^{(l)}(x,x) = 1. \tag{B-38}$$

In the case of $CNTK$ for the step $l$, $\Sigma_{p,q}^{(l)}(x,x)$

$$\boldsymbol{\Sigma}_{p,q}^{(l)}(x,x) = \frac{\lambda(\pi - arccos(\lambda)) + \sqrt{1-\lambda^2}}{\pi} \cdot c_1 c_2, \ \dot{\boldsymbol{\Sigma}}_{p,q}^{(l)}(x,x) = \frac{\pi - arccos(\lambda)}{\pi}. \tag{B-39}$$

In which $\boldsymbol{D\Lambda D} = \begin{pmatrix} \boldsymbol{\Sigma}_{p,p}^{(l-1)}(x,x) & \boldsymbol{\Sigma}_{p,q}^{(l-1)}(x,x) \\ \boldsymbol{\Sigma}_{q,p}^{(l-1)}(x,x) & \boldsymbol{\Sigma}_{q,q}^{(l-1)}(x,x) \end{pmatrix}$, thus $c_1 c_2 = \sqrt{\boldsymbol{\Sigma}_{p,p}^{(l-1)}(x,x)\boldsymbol{\Sigma}_{q,q}^{(l-1)}(x,x)}$ and $\lambda = \frac{\boldsymbol{\Sigma}_{p,q}^{(l-1)}(x,x)}{c_1 c_2}$. $\square$

similar conclusion can be obtained to the case in the CNTK, in which $\mathbb{E}(\lambda) \rightarrow \dfrac{\mathbb{E}\left(\boldsymbol{\Sigma}_{p,q}^{(l-1)}(x,x')\right)}{\sqrt{\mathbb{E}\left(\boldsymbol{\Sigma}_{p,p}^{(l-1)}(x,x)\right)\mathbb{E}\left(\boldsymbol{\Sigma}_{q,q}^{(l-1)}(x,x)\right)}}$ and $\mathbb{D}(\lambda) \rightarrow 0$ along with $n_l \rightarrow \infty$. While there exists a specific case in the step $l = 1$ due to $n_0$ is a constant that is differ from $n_k(0 < k < L)$. When $l \geq 1$, in the NTK:

$$
\begin{aligned}
\mathbb{E}\left(\Sigma^{(l)}(x,x')\right) &= \mathbb{E}\left(\frac{\lambda(\pi - arccos(\lambda)) + \sqrt{1 - \lambda^2}}{\pi} \cdot c_1 c_2\right) \\
&\leq \mathbb{E}\left(\frac{\lambda\pi + 1}{\pi} \cdot c_1 c_2\right) \\
&= \mathbb{E}\left(\Sigma^{(l-1)}(x,x')\right) + \frac{1}{\pi}\mathbb{E}\left(\sqrt{\Sigma^{(l-1)}(x,x)\Sigma^{(l-1)}(x',x')}\right) \\
&\leq \mathbb{E}\left(\Sigma^{(l-1)}(x,x')\right) + \frac{1}{\pi}\mathbb{E}\left(\Sigma^{(l-1)}(x,x)\right)
\end{aligned}
\tag{B-40}
$$

in which the last step can be splited to two steps as follows by Jensen inequality (since the function $g(x) = \sqrt{x}$ is concave):

$$
\begin{aligned}
\mathbb{E}\left(\sqrt{\Sigma^{(l-1)}(x,x)\Sigma^{(l-1)}(x',x')}\right) &= \mathbb{E}\left(\Sigma^{(l-1)}(x,x)\right), \quad x = x' \\
\mathbb{E}\left(\sqrt{\Sigma^{(l-1)}(x,x)\Sigma^{(l-1)}(x',x')}\right) &= \mathbb{E}\left(\sqrt{\Sigma^{(l-1)}(x,x)}\right)\mathbb{E}\left(\sqrt{\Sigma^{(l-1)}(x',x')}\right) \\
&\leq \sqrt{\mathbb{E}\left(\Sigma^{(l-1)}(x,x)\right)}\sqrt{\mathbb{E}\left(\Sigma^{(l-1)}(x',x')\right)} \\
&= \mathbb{E}\left(\Sigma^{(l-1)}(x,x)\right), x \neq x'
\end{aligned}
\tag{B-41}
$$

similar to the CNTK:

$$
\begin{aligned}
&\mathbb{E}\left(\boldsymbol{\Sigma}_{p,q}^{(l)}(x,x')\right) = \mathbb{E}\left(\frac{\lambda(\pi - arccos(\lambda)) + \sqrt{1 - \lambda^2}}{\pi} \cdot c_1 c_2\right) \\
&\leq \mathbb{E}\left(\boldsymbol{\Sigma}_{\{p,q\}}^{(l-1)}(x,x')\right) + \frac{1}{\pi}\mathbb{E}\left(\sqrt{\boldsymbol{\Sigma}_{p,p}^{(l)}(x,x)\boldsymbol{\Sigma}_{q,q}^{(l)}(x',x')}\right) \\
&\leq \begin{cases} \mathbb{E}\left(\boldsymbol{\Sigma}_{\{p,q\}}^{(l-1)}(x,x')\right) + \frac{1}{\pi}\sqrt{\mathbb{E}\left(\boldsymbol{\Sigma}_{p,p}^{(l-1)}(x,x)\right)\mathbb{E}\left(\boldsymbol{\Sigma}_{q,q}^{(l-1)}(x',x')\right)}, & x \neq x' \\ \frac{2+\pi}{\pi}\mathbb{E}\left(\boldsymbol{\Sigma}_{p,q}^{(l-1)}(x,x)\right) + \frac{1}{\pi}\sqrt{\mathbb{E}\left(\boldsymbol{\Sigma}_{p,p}^{(l-1)}(x,x)\right)\mathbb{E}\left(\boldsymbol{\Sigma}_{q,q}^{(l-1)}(x,x)\right)}, & x = x' \end{cases}
\end{aligned}
\tag{B-42}
.
$$

$$
\mathbb{E}\left(\dot{\boldsymbol{\Sigma}}_{p,q}^{(l)}(x,x')\right) = \mathbb{E}\left(\frac{\pi - arccos(\lambda)}{\pi}\right) \leq 1, \quad \mathbb{E}\left(\Sigma^{(l)}(x,x')\right) = \mathbb{E}\left(\frac{\pi - arccos(\lambda)}{\pi}\right) \leq 1
\tag{B-43}
$$

Now we induce the twice:

$$
\begin{aligned}
\mathbb{E}\left(\Sigma^{2(l)}(x,x')\right) &= \mathbb{E}\left(\left(\frac{\lambda(\pi - arccos(\lambda)) + \sqrt{1 - \lambda^2}}{\pi}\right)^2 \cdot c_1^2 c_2^2\right) \\
&\leq \mathbb{E}\left(\left(\frac{\pi^2 - 1}{\pi^2}\lambda^2 + \frac{2}{\pi}\lambda\sqrt{1 - \lambda^2} + \frac{1}{\pi^2}\right) \cdot c_1^2 c_2^2\right) \\
&= \frac{\pi^2 - 1}{\pi^2}\mathbb{E}\left((\Sigma^{(l-1)})^2(x,x')\right) + \frac{2\pi\lambda\sqrt{1 - \lambda^2} + 1}{\pi^2}\mathbb{E}\left(\Sigma^{(l-1)}(x,x)\Sigma^{(l-1)}(x',x')\right) \\
&\leq \frac{\pi^2 - 1}{\pi^2}\mathbb{E}\left((\Sigma^{(l-1)})^2(x,x')\right) + \frac{\pi + 1}{\pi^2}\mathbb{E}\left(\Sigma^{(l-1)}(x,x)\Sigma^{(l-1)}(x',x')\right) \\
&= \begin{cases} \frac{\pi^2-1}{\pi^2}\mathbb{E}\left((\Sigma^{(l-1)})^2(x,x')\right) + \frac{\pi+1}{\pi^2}\mathbb{E}\left(\Sigma^{(l-1)}(x,x)\right)\mathbb{E}\left(\Sigma^{(l-1)}(x',x')\right), & x \neq x' \\ \frac{\pi+1}{\pi}\mathbb{E}\left((\Sigma^{(l-1)})^2(x,x')\right), & x = x' \end{cases}
\end{aligned}
\tag{B-44}
.
$$

Table 3: Performance on ModelNet10$_3$ of Kernel Regression (KR) and Supervised Learning (SL). All supervised networks are of the vanilla structure. GAP indicates global average pooling is added to the final layer. The exponential parameter of the Gaussian kernel is 1.

|     | Model | D = 5 | D = 10 |
|-----|-------|-------|--------|
| SL | 1dCNN | 42.18 | 21.70 |
|    | 1dCNN_GAP | 39.54 | 21.26 |
| KR | 1dCNTK | 44.38 | 43.39 |
|    | 1dCNTK_GAP | 76.87 | 76.54 |
|    | NTK | 19.98 | 14.54 |
|    | Gaussian Kernel | 21.15 | – |

similar to the CNTK:

$$
\mathbb{E}\left(\mathbf{\Sigma}_{p,q}^{2(l)}(x,x')\right) = \mathbb{E}\left(\left(\frac{\lambda(\pi - arccos(\lambda)) + \sqrt{1-\lambda^2}}{\pi}\right)^2 \cdot c_1^2 c_2^2\right)
$$

$$
\leq \frac{\pi^2 - 1}{\pi^2}\mathbb{E}\left((\mathbf{\Sigma}_{p,q}^{(l-1)})^2(x,x')\right) + \frac{\pi + 1}{\pi^2}\mathbb{E}\left(\mathbf{\Sigma}_{p,p}^{(l-1)}(x,x)\mathbf{\Sigma}_{q,q}^{(l-1)}(x',x')\right)
$$

$$
\leq \begin{cases} \frac{\pi^2-1}{\pi^2}\mathbb{E}\left((\mathbf{\Sigma}_{p,q}^{(l-1)})^2(x,x')\right) + \frac{\pi+1}{\pi^2}\mathbb{E}\left(\mathbf{\Sigma}_{p,p}^{(l-1)}(x,x)\right)\mathbb{E}\left(\mathbf{\Sigma}_{q,q}^{(l-1)}(x',x')\right), & x \neq x' \\ \frac{(\pi+1)^2}{\pi^2}\mathbb{E}\left((\mathbf{\Sigma}_{p,q}^{(l-1)})^2(x,x')\right) + \frac{\pi+1}{\pi^2}\mathbb{E}\left(\mathbf{\Sigma}_{p,p}^{(l-1)}(x,x)\right)\mathbb{E}\left(\mathbf{\Sigma}_{q,q}^{(l-1)}(x,x)\right), & x = x' \end{cases}.
$$

(B-45)

# E  EXTRA EXPERIMENTS

## E.1  VANILLA METHODS FOR POINTCLOUD

Based on the kernel regression, we tested NTK, 1dCNTK, and 1dCNTK with a global average pooling layer 1dCNTK$_{GAP}$ on ModelNet10$_3$. Additionally, we conducted experiments using a traditional Gaussian kernel for comparison. Furthermore, we trained the corresponding vanilla neural network structures based on these kernels.

The results, shown in Table 1, indicate the following:

- Vanilla networks without empirical adjustments perform poorly on point cloud data, with performance degrading as network depth increases.
- The NTK corresponding to MLPs fails to perform effective kernel regression on point cloud data, performing even worse than the Gaussian kernel.
- 1dCNTK is effective kernel regression for point cloud data, and 1dCNTK with a global average pooling significantly outperforms 1dCNTK without a pooling.
- All kernels show no substantial improvement with increasing depth.

