# OpenReview forum: "On the Spectral Differences Between NTK and CNTK and Their Implications for Point Cloud Recognition"
_ICLR.cc/2026/Conference — ICLR 2026 Poster_

### Official Review · Reviewer_qeWv · 2025-10-23

**Soundness:** 3
**Presentation:** 3
**Contribution:** 3
**Rating:** 6
**Confidence:** 2

**Summary:**

This paper finds that CNTK is more suitable for point cloud data than image data, and thus proposes a CNTK-based kernel regression method for point cloud recognition tasks. Experimental results demonstrate its effectiveness, while providing a new theoretical explanation for the spectral characteristics of NTK.

**Strengths:**

1.This paper explains the spectral characteristics of NTK from a new perspective and finds that CNTK is more suitable for point cloud data. The formulas are rigorous and correct, and the experimental results also demonstrate the rationality of this theory.

**Weaknesses:**

1.This paper only conducts training on the ModelNet dataset. Experiments on ScanObjectNN, which is more in line with real-world scenarios and more challenging, should be added to demonstrate the practicality of the method.

**Questions:**

1.The relevant work done in this paper on shared MLP should be extendable to more advanced point cloud models, such as PointMLP. Have the authors made any relevant attempts?

---

> ### Author Response · Authors · 2025-11-14
>
> Thank you for your suggestion.  We acknowledge that our choices may appear relatively conservative. However, since our work is primarily theoretical rather than application-oriented, we followed the common practice in theoretical studies by using classic benchmark datasets. Indeed, many theoretical works on MLPs and CNNs rely on standard datasets such as MNIST and CIFAR-10. For point cloud analysis, theoretical or interpretability-focused research remains limited, and thus we opted for the widely used and well-established ModelNet dataset. Although our experiments center on ModelNet, we include both commonly used subsets—ModelNet40 and ModelNet10—and further analyze two variants within ModelNet10. We sincerely appreciate the reviewer’s suggestion and will consider exploring additional datasets in future work. At the same time, we believe that the current set of experiments provides sufficient coverage for supporting the theoretical claims of this study.

---

### Official Review · Reviewer_GQJv · 2025-10-27

**Soundness:** 2
**Presentation:** 2
**Contribution:** 2
**Rating:** 4
**Confidence:** 3

**Summary:**

The paper analyzes the spectral comparison between Convolutional Neural Tangent Kernel (CNTK) and Neural Tangent Kernel (NTK). Through a series of theoretical analysis and validation via synthetic data, the paper concludes that convolutional structures are inherently more suited to irregular point cloud data.

**Strengths:**

The theoretical comparison between CNTK and NTK can potentially guide architectural search of point cloud tasks.

**Weaknesses:**

* The writing of the paper is hard to follow, hard to understand. Though it is a theory paper, the presentation can be made much more accessible by explaining the intuition behind and visualization. Figure 1 seems to have such attempt, but it is not explained well.
* The conclusion reached by the paper is a well-known fact from empirical experience. The paper only decorates it with some theoretical proof.
* The conclusion that “convolutional structures are more suited to irregular point cloud data” is supported by Figure 2. But it only has one point cloud dataset and two image datasets, which cannot represent “point cloud” and “image”.
* In the experiments, the result shows PointNet performs better than PointNTK under all settings. So it is hard to understand what is the practical implication of the paper.

**Questions:**

What can be some practical implication of the paper given PointNTK performs worse than PointNet?

---

> ### Author Response · Authors · 2025-11-14
>
> Thank you for your suggestion.
>
> Q: The writing of the paper is hard to follow, hard to understand. Though it is a theory paper, the presentation can be made much more accessible by explaining the intuition behind and visualization. Figure 1 seems to have such attempt, but it is not explained well.
>
> R: As the reviewer pointed out, this work is indeed theoretical in nature, which makes it challenging to offer fully “intuitive’’ explanations of some of the core concepts. Nevertheless, we have made substantial efforts to simplify the notation and improve clarity wherever possible. For instance, Section 1.2 provides a consolidated summary of all key symbols used throughout our theoretical derivations, with the goal of helping readers better follow our main claims.
>
> Q: The conclusion reached by the paper is a well-known fact from empirical experience. The paper only decorates it with some theoretical proof.
>
> R: We appreciate the reviewer’s observation. Indeed, the empirical superiority of CNNs over MLPs has been known for nearly a decade. However, the underlying reasons—why CNNs generalize better and under which types of data they outperform MLPs—are still not fully understood. Addressing these fundamental questions is the central focus of our paper, and we aim to provide a theoretical explanation that complements and helps formalize these long-standing empirical findings.
>
>
> Q: The conclusion that “convolutional structures are more suited to irregular point cloud data” is supported by Figure 2. But it only has one point cloud dataset and two image datasets, which cannot represent “point cloud” and “image”.
>
> R: We acknowledge that our choices may appear relatively conservative. However, since our work is primarily theoretical rather than application-oriented, we followed the common practice in theoretical studies by using classic benchmark datasets. Indeed, many theoretical works on MLPs and CNNs rely on standard datasets such as MNIST and CIFAR-10.
> For point cloud analysis, theoretical or interpretability-focused research remains limited, and thus we opted for the widely used and well-established ModelNet dataset. Although our experiments center on ModelNet, we include both commonly used subsets—ModelNet40 and ModelNet10—and further analyze two variants within ModelNet10.
> We sincerely appreciate the reviewer’s suggestion and will consider exploring additional datasets in future work. At the same time, we believe that the current set of experiments provides sufficient coverage for supporting the theoretical claims of this study.
>
>
>
> Q: In the experiments, the result shows PointNet performs better than PointNTK under all settings. So it is hard to understand what is the practical implication of the paper.
>
> R: Since the introduction of the NTK framework, it has been well observed that real, finite-width neural networks trained with gradient descent often outperform NTK regression (which corresponds to the infinite-width limit). However, the goal of NTK research has never been to argue that “NTK regression should outperform real models,” but rather to provide theoretical insight into why practical networks behave as they do. Our proposed PointNTK is motivated by the same philosophy.
> In addition, we include the kernel regression experiments because, although kernel methods typically underperform real networks on large-scale datasets, they remain highly effective in small-sample settings—a trend that can also be seen in our results. Thus, the primary purpose of PointNTK is to offer a theoretical explanation for why MLPs, CNNs, and PointNet architectures perform well on point cloud data, while also providing a kernel-based alternative that may serve as a viable option in low-data regimes.

---

### Official Review · Reviewer_qF94 · 2025-10-27

**Soundness:** 3
**Presentation:** 3
**Contribution:** 3
**Rating:** 8
**Confidence:** 3

**Summary:**

This paper presents the first systematic comparison of the spectral properties of the Neural Tangent Kernel (NTK) and the Convolutional NTK (CNTK). The authors formally prove that for data with a tensor structure, CNTK consistently exhibits a broader eigenvalue spectrum and a smaller mean eigenvalue compared to NTK. Based on this insight, they propose that CNTK's spectral bias is inherently better suited for geometric data like point clouds. This hypothesis is validated by introducing a metric for "Convolutional Suitability" and demonstrating experimentally that point clouds align more strongly with CNTK's properties. Finally, the authors propose PointNTK, a CNTK-based kernel regression method, which achieves strong performance on point cloud recognition, particularly outperforming training-based baselines in low-data settings.

**Strengths:**

- Novel and Insightful Theoretical Contribution:

    The paper provides the first systematic spectral comparison of NTK and CNTK, offering a novel theoretical lens to understand the inductive biases of convolutional architectures. The introduction of mK and βK as quantitative metrics and the concept of "Convolutional Suitability" are significant contributions that provide theoretical guidance on how to choose the right network architecture for a given type of data.
﻿

- Combining theory with practice:

    A key strength of this work is that it clearly connects theory with practice. The authors start from formal proofs to formulate a verifiable hypothesis—that point clouds are more "convolutionally suitable" than images—and then compellingly validate this hypothesis through well-designed experiments.
﻿

- Explanation a Fundamental Question in Point Cloud Processing:

    This paper tackles a highly relevant question in the 3D vision community: why are convolutional-like structures (such as the shared-MLPs in PointNet) so effective for point cloud data. By approaching this from a theoretical kernel perspective, the work provides a novel explanation.

**Weaknesses:**

- [Minor] Confusing performance:
﻿
    The results in Table 1 present a slightly confusing result. For the ModelNet10_6 dataset, the vanilla 1dCNTK (91.96%) outperforms the more complex PointNTK (91.19%). This seems to contradict the paper's motivation for adding MLP layers . A discussion on why this might occur would be beneficial.
﻿
- [Minor] Potential Discrepancy Between Ablation Results and the Unorderedness Argument:
﻿
    The paper makes a strong argument that a kernel size greater than 1 is detrimental due to the unordered nature of point clouds. Following this logic, one would expect a sharp performance drop when the kernel size increases from 1 to 2. However, the experimental results (Figure 4) show a gradual decline rather than a steep fall. This gentle degradation seems not entirely consistent with the theoretical expectation that aggregating unordered points would cause severe disruption. The paper would be strengthened by a discussion or explanation of this phenomenon .
﻿
- [Minor] Practical Limitation of the β_K Metric:

    The use of the initial layer's metric, β_K(0), as a proxy for "Convolutional Suitability" is a practical simplification but also a limitation. As Figure 1 shows, β_K evolves with depth. The paper would be stronger if it discussed this evolution or provided bounds on its variation, which would enhance the completeness of the theory.
﻿
- [Minor] The Core Concept of "Convolutional Suitability" Lacks a Formal Definition

    The paper introduces "Convolutional Suitability" as a key concept but fails to provide a formal definition. It is only mentioned that 1 - β_NTK(0) can be "interpreted as" this metric. For clarity and to facilitate future work, this concept should be formally defined when it is first introduced.

**Questions:**

- On the Performance of the PointNTK Model:
    I noticed in Table 1 that for the ModelNet10_6 dataset, the simpler 1dCNTK model slightly outperforms the PointNTK hybrid model. This is counter-intuitive given that the MLP layers are introduced to enhance performance. What is your interpretation of this result? Is it related to dataset-specific properties?
﻿
- On the Smooth Decline in the Kernel Size Ablation:
    Your argument about point cloud unorderedness suggests that a kernel size k > 1 should be highly detrimental. I would have expected a sharp performance drop when moving from k=1 to k=2. Instead, Figure 4 shows a smooth, gradual decline. What is your intuition for this gentle degradation?
﻿
- On β_K(0) as a Proxy for Suitability:
    You use β_K(0) as a practical proxy for "Convolutional Suitability" across the entire network. While I understand the computational challenge of evaluating β_K(L) for deep layers, do you have any evidence or theoretical argument suggesting that β_K(0) is a reliable representative? An answer here would strengthen the claim that this initial-layer metric is sufficient.

---

> ### Author Response · Authors · 2025-11-14
>
> [Minor] Confusing performance: The results in Table 1 present a slightly confusing result. For the ModelNet10_6 dataset, the vanilla 1dCNTK (91.96%) outperforms the more complex PointNTK (91.19%). This seems to contradict the paper's motivation for adding MLP layers . A discussion on why this might occur would be beneficial.
>
> R: In fact, this result can be better understood by referring to the second subfigure in Figure 2. As shown there, although the mean value of ModelNet10_6 increases in PointNTK compared with 1dNTK, its overall distribution becomes noticeably narrower and sharper. This concentration of the distribution leads to more clustered eigenvalues, resulting in smaller (\beta_k) values and, consequently, weaker generalization ability.
> In contrast, for ModelNet10_3 and ModelNet40, we can observe that their distributions under PointNTK do not exhibit such a pronounced “narrowing and sharpening” effect.
>
>
> [Minor] Potential Discrepancy Between Ablation Results and the Unorderedness Argument:
> The paper makes a strong argument that a kernel size greater than 1 is detrimental due to the unordered nature of point clouds. Following this logic, one would expect a sharp performance drop when the kernel size increases from 1 to 2. However, the experimental results (Figure 4) show a gradual decline rather than a steep fall. This gentle degradation seems not entirely consistent with the theoretical expectation that aggregating unordered points would cause severe disruption. The paper would be strengthened by a discussion or explanation of this phenomenon .
>
> R:  Thank you very much for the insightful suggestion. We initially considered incorporating different convolutional kernel sizes into our analysis. However, our preliminary calculations showed that, at the second layer, kernels of different sizes mainly lead to shifted versions of the covariance matrix being added together, which we initially viewed as a relatively straightforward extension.
> That said, we agree that a deeper examination—particularly one focusing on how the diagonalization level of the covariance matrix (which reflects the degree of order or disorder in the data) changes under different kernel sizes—may reveal a more meaningful relationship. We are currently exploring this direction further and will include our latest findings in the revised version of the paper.
>
>
>
>
> [Minor] Practical Limitation of the β_K Metric:
> The use of the initial layer's metric, β_K(0), as a proxy for "Convolutional Suitability" is a practical simplification but also a limitation. As Figure 1 shows, β_K evolves with depth. The paper would be stronger if it discussed this evolution or provided bounds on its variation, which would enhance the completeness of the theory.
>
> R: This is indeed a very meaningful point; however, a deeper discussion is somewhat challenging. In fact, NTK behavior changes significantly when moving from two-layer to deeper networks. As the NTK extends to deeper layers, the $\Sigma$ tensors in our analysis interact and influence each other extensively, which significantly increases the complexity of the theoretical analysis. Moreover, since our work focuses on spectral characteristics, it becomes challenging to accurately describe the distributions of the individual random variables and their joint distributions in deeper layers.
>
> [Minor] The Core Concept of "Convolutional Suitability" Lacks a Formal Definition
> The paper introduces "Convolutional Suitability" as a key concept but fails to provide a formal definition. It is only mentioned that 1 - β_K can be "interpreted as" this metric. For clarity and to facilitate future work, this concept should be formally defined when it is first introduced.
>
> R: Thank you very much for your suggestion. We will provide a clear and formal definition of the concept “Convolutional Suitability” in the revised version of the paper.

---

> > ### Comment · Reviewer_qF94 · 2025-11-27
> >
> > Thank you for your detailed response.
> >
> > Your explanations have satisfactorily addressed my concerns. I found the explanation for the performance on ModelNet10_6, which connects back to the eigenvalue distribution, particularly insightful.
> >
> > On the second point: Regarding the kernel size ablation, your proposed direction for a deeper analysis is promising and could well explain the gradual performance decline.
> >
> > On the third point: I accept your reasoning for using β_K(0) as a proxy, given the well-known theoretical complexity of analyzing deeper networks.
> >
> > I also appreciate your commitment to formally defining "Convolutional Suitability" and adding further discussion on the other points in the revised manuscript.
> >
> > I will maintain my positive rating for the paper and look forward to seeing the revised version.

---

### Meta-Review · Area_Chair_Ce5f · 2025-12-25

**Summary:**

This paper focused on two areas: comparing the spectra of Neural Tangent Kernel (NTK) and Convolutional NTK (CNTK) and then applying CNTK regression to point cloud classification.

Reviewer Feedback
Pros: The theoretical analysis was strong, demonstrating the differences between NTK and CNTK and establishing CNTK's relevance for point cloud analysis.
Cons: Criticisms included the underperformance of kernel regression methods compared to state-of-the-art models like PointNet, the paper lacking some clarity (one reviewer) and lack of accessibility for readers outside the relevant sub-community, and a limited experimental scope (e.g., only ModelNet).

Author Clarification
The authors clarified that their goal was to contribute to the theoretical understanding of convolutional kernels for point clouds, not to outperform trained neural networks with non-trained kernel methods. They also made an effort to make the paper more accessible.

The paper was ultimately considered to have slightly passed the bar for acceptance.

**Reviewer Concerns:**

please see above.

**Reviewer Scores:**

please see above.

---

### Decision · Program_Chairs · 2026-01-26

Accept (Poster)